# Next Point-of-Interest Recommendation Based on Joint Mining of Spatial–Temporal and Semantic Sequential Patterns

**Jing Tian** [1], **Zilin Zhao** [1] **and Zhiming Ding** [2,*]

1   Faculty of Information Technology, Beijing University of Technology, Beijing 100124, China;
    tianj@emails.bjut.edu.cn (J.T.); zhaozilin@emails.bjut.edu.cn (Z.Z.)
2   The Institute of Software, Chinese Academy of Sciences, Beijing 100190, China
*   Correspondence: zhiming@iscas.ac.cn

**Abstract:** With the widespread use of the location-based social networks (LBSNs), the next point-of-interest (POI) recommendation has become an essential service, which aims to understand the user's check-in behavior at the current moment by analyzing and mining the correlations between the user's check-in behaviors within his/her historical trajectory and then recommending the POI that the user is most likely to visit at the next time step. However, the user's check-in trajectory presents extremely irregular sequential patterns, such as spatial–temporal patterns, semantic patterns, etc. Intuitively, the user's visiting behavior is often accompanied by a certain purpose, which makes the check-in data in LBSNs often have rich semantic activity characteristics. However, existing research mainly focuses on exploring the spatial–temporal sequential patterns and lacks the mining of semantic information within the trajectory, so it is difficult to capture the user's visiting intention. In this paper, we propose a self-attention- and multi-task-based method, called MSAN, to explore spatial–temporal and semantic sequential patterns simultaneously. Specifically, the MSAN proposes to mine the user's visiting intention from his/her semantic sequence and uses the user's visiting intention prediction task as the auxiliary task of the next POI recommendation task. The user's visiting intention prediction uses hierarchical POI category attributes to describe the user's visiting intention and designs a hierarchical semantic encoder (HSE) to encode the hierarchical intention features. Moreover, a self-attention-based hierarchical intention-aware module (HIAM) is proposed to mine temporal and hierarchical intention features. The next POI recommendation uses the self-attention-based spatial–temporal-aware module (STAM) to mine the spatial–temporal sequential patterns within the user's check-in trajectory and fuses this with the hierarchical intention patterns to generate the next POI list. Experiments based on two real datasets verified the effectiveness of the model.

**Keywords:** next POI recommendation; intention prediction; self-attention network; multi-task learning

## 1. Introduction

With the rapid development of Internet technology and the widespread application of mobile smart devices with wireless positioning functions, location-based social network (LBSN) services are gradually emerging [1]. At present, the application software based on LBSNs mainly includes the foreign Gowalla, Yelp, Foursquare, etc., as well as the domestic Meituan, Dianping, and Weibo. According to Foursquare's statistics, more than 550 million users visit the Foursquare website every month, and the total number of check-ins exceeds three billion times per month. As shown in Figure 1, users can perform location check-ins, information sharing, and online social activities anytime and anywhere, resulting in massive spatial–temporal trajectory data and social activity data. In return, these check-in data give LBSNs the chance to create more-individualized services, such as point-of-interest (POI) recommendations.

Traditional POI recommendation aims to recommend the POIs that the user is interested in based on his/her historical check-in records, which treats the user's historical trajectory as a set, regardless of the time order. Compared with the traditional POI recommendation, the next POI recommendation focuses on mining the sequential patterns within the user's check-in trajectory and aims to recommend the POI that the user is most likely to visit at the next time step. Most of the existing models focus on mining the spatial–temporal sequential patterns within the user's check-in trajectories [2–5], which mainly include sequential transition patterns and spatial–temporal correlations (e.g., spatial–temporal interval features) between POIs in the trajectories. Early models were mainly based on the Markov chain (MC). However, these models only considered sequential transition patterns. Later, many studies [2,5,6] extended recurrent neural networks (RNNs) to incorporate the spatial–temporal features within the user's check-in trajectory.

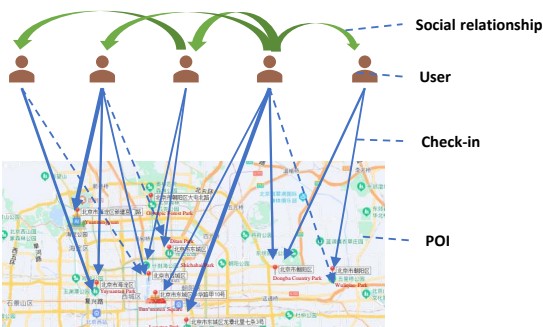

**Figure 1.** This is a typical LBSN system (the line weight indicates the user's check-in frequency, and the greater the weight of the line, the higher the user's check-in frequency at the POI).

Recently, many methods [3,4,7] have modeled the spatial–temporal sequential patterns within the user's check-in trajectory based on a self-attention mechanism and have achieved remarkable results. GeoSAN [3] models geographic information by partitioning geographic space into hierarchical grids and uses self-attention mechanisms to learn geographic correlations between POIs in trajectory sequences. The STAN model [4] proposes to directly integrate the time intervals and geographical distances between each check-in behavior and aggregates all the relevant visits from the user's check-in trajectory to recall the candidates with the highest probability.

However, the user's check-in behavior presents extremely irregular sequential patterns owing to multiple factors such as temporal correlations, geographical correlations, and semantic correlations. Existing methods mainly focus on exploring the spatial–temporal sequential patterns and lack the mining of semantic information within the user's check-in sequence, so it is difficult to accurately capture users' visiting intentions. Intuitively, the user's visiting behavior is often accompanied by a certain purpose. For example, if a user wants to go shopping, he/she may visit a department store. Therefore, mining the user's visiting intention from the semantic trajectory and combining the prediction of the user's visiting intention with the next POI recommendation can further improve the recommendation accuracy of the model. However, few studies have paid attention to it at present.

In this paper, we propose a novel next POI recommendation method, called the MSAN, to explore spatial–temporal and semantic sequential patterns simultaneously. This model proposes to mine the user's visiting intention from the semantic sequence and uses the user's visiting intention prediction task as the auxiliary task of the next POI recommendation. By building a multi-task learning framework to achieve knowledge sharing between the two tasks, the performance and generalization ability of the model are improved. The user's visiting intention prediction task uses hierarchical POI category attributes to describe the user's visiting intention and designs a hierarchical semantic encoder (HSE) to encode the hierarchical intention features. Moreover, it proposes a hierarchical intention-

aware module (HIAM) based on the self-attention to capture temporal and hierarchical intention features. The next POI recommendation uses the self-attention-based spatial–temporal-aware module (STAM) to mine the spatial–temporal sequential features within the user's check-in trajectory and fuses this with the hierarchical intention features to generate the next POI list. The main contributions of this paper are listed as follows:

- To improve the performance and generalization of recommendation, we propose a self-attention- and multi-task-learning-based method (MSAN) to fully explore spatial–temporal and semantic sequential patterns simultaneously, which mines the user's visiting intention from his/her semantic sequence and uses the user's visiting intention prediction task as the auxiliary task of the next POI recommendation task.
- We propose to use hierarchical POI category attributes to describe the user's visiting intention. To mine the user's visiting intention, we designed a hierarchical semantic encoder (HSE) to encode the hierarchical intention feature. Moreover, a self-attention-based hierarchical intention-aware module (HIAM) is proposed to consider the temporal effect for aggregating relevant POI categories within the user's semantic sequence to update the intention representation of each check-in.
- Experiments based on two real-world datasets demonstrated that the MSAN model outperformed most of the current state-of-the-art baseline models. Thus, we verified the effectiveness of using the visiting intention prediction task as the auxiliary task of the next POI recommendation.

The current research status of POI recommendation is listed in the Section 1. The Section 2 will briefly review related work on POI recommendation. In the Section 3, we introduce some preliminary work. The Section 4 details our proposed MSAN method for the next POI recommendation. In the Section 5, a series of experiments on two real datasets is conducted and the experimental results' analyses are given to verify the effectiveness of the model. The Section 6 summarizes the content of this paper.

## 2. Related Work

In this section, we conduct a literature review on traditional POI recommendation and the next POI recommendation.

### 2.1. Traditional POI Recommendation

Traditional POI recommendation aims to recommend the POIs that the user is interested in. In order to improve the accuracy of POI recommendation, related works tried to mine the temporal correlations, spatial correlations, social correlations, and semantic correlations to construct recommendation models.

Temporal modeling: The time influence in POI recommendation is mainly reflected in three aspects: periodicity, continuity, and inconsistency [8]. Most of the studies [9–13] modeled the periodicity and continuity of time influence. Li et al. [10] considered that the user's check-in behavior presents daily patterns, so they divided the time of the day into time slices in units of hours and combined the tensor factorization (TF) model to capture the category preferences of users in different time periods. Wang et al. [13] considered the weekly pattern characteristics of the user's check-in behavior, so they divided the time of the week into two patterns: weekdays and weekends, and combined matrix factorization (MF) to recommend POIs to users. Temporal continuity means that the POIs where users check-in in continuous time are potentially relevant, which is usually modeled in the next POI recommendation.

Geographical modeling: In LBSNs, the user's check-in behavior presents the spatial clustering phenomenon [3]. Ye et al. [14] proposed that the geographical distance between the POIs that users continuously visit presents a power-law distribution and modeled geographical factors based on the power-law distribution. Furthermore, Cheng et al. [15] considered that the spatial distribution of the user's check-in behavior presents a multi-centered Gaussian distribution, and modeling based on the one-dimensional geographical distance alone cannot capture the two-dimensional multi-centered Gaussian distribution, so

they proposed a personalized multi-centered Gaussian distribution model (MGM) to model geographic patterns and incorporate these into the MF model for POI recommendation.

Social modeling: In LBSNs, users tend to visit POIs that their friends have visited [16]. Most of the related works extracted the similarity between users from their social relationship and integrated this into the traditional memory-based or model-based collaborative filtering (CF) to improve the recommendation performance [17–19]. Other studies [20,21] fused social factors as regularization items or the weights of latent factor models.

Semantic modeling: The category information of POI can reflect the user's activity theme. Most researchers focused on the category information of the POI to analyze the user's check-in behavior from the perspective of semantics. For example, a user has visited a supermarket, which means that he/she is going shopping there. Existing research mainly modeled the category information of the POI from the following two perspectives. On the one hand, different users have different category preferences; on the other hand, the same user has different category preferences at different times. Bao et al. [22] integrated the user's category preference into the CF method and calculated the similarity between different users by calculating the user's POI category deviation. Liu et al. [23] clustered users according to their POI category attributes based on their historical check-in records and replaced the user–POI matrix by constructing a user–category matrix and then used MF technology to predict the top-$N$ category list.

## 2.2. Next POI Recommendation

The next POI recommendation aims to mine the sequential patterns within users' check-in trajectories and recommend the POIs that the users most likely visit at the next time step.

Most of the existing models focus on mining the spatial–temporal sequential features within the user's check-in trajectories [2–5], which mainly include sequential transition patterns and spatial–temporal correlations (e.g., spatial–temporal interval features) between POIs within the trajectories. Early works mainly relied on the Markov chain (MC) to model the sequential patterns. Zhang et al. [24] proposed an additive MC model to learn the check-in probability between two consecutive check-in behaviors. However, Markov-based models mainly focus on the transition probability between two consecutive visits and neglect mining high-order transition patterns. To model the high-order sequential patterns, RNN-based models are proposed. Meanwhile, researchers also exploit temporal and spatial correlations to assist sequential recommendation. The STRNN model [2] introduces the temporal intervals and spatial distances between consecutive check-in behaviors to capture the spatial–temporal correlations between them, which was the first work to mine the spatial–temporal correlations within the user's check-in sequence. The HST-LSTM model [25] integrates spatial–temporal context information into LSTM and proposes a trajectory hierarchy division method based on functional areas, which alleviates the problem of data sparsity by mining the spatial–temporal transition characteristics between different functional areas. However, due to the special structural design of RNNs, these RNN-based models are limited to modeling long-term sequential dependencies [26].

In recent years, the self-attention mechanism has been shown to be powerful in modeling long sequential patterns [27]. In view of this, the current state-of-the-art models [3,4,7] are all based on the self-attention mechanism to model the spatial–temporal sequence characteristics within the user's check-in trajectories and have achieved remarkable results. GeoSAN [3] mines geographic information by partitioning geographic space into hierarchical grids and uses the self-attention mechanism to learn geographic correlations between POIs in trajectory sequences. The STAN model [4] proposes to directly integrate the time intervals and geographical distances between each check-in behavior and aggregate all the relevant visits from the user's check-in trajectory to recall the candidates with the most probability. Although all the above models have achieved remarkable results in the mining of spatial–temporal sequential features, they all neglect the exploration of semantic information within the user's check-in sequence. If the user's visiting intention

can be understood from the user's semantic activity characteristics, the recommendation performance of the model can be further improved.

## 3. Motivation and Preliminaries

### 3.1. Motivation

The user's check-in behavior usually has a certain purpose, and the POIs frequently visited by users in LBSNs also show corresponding functional attributes. Therefore, the check-in trajectory carries rich semantic information. By capturing the hidden semantic factors in the trajectory sequence, we can better understand the user's visiting intention and achieve more-accurate predictions. However, in the research of the next POI recommendation, most studies only mine user check-in sequence patterns from time and geographical features, ignoring the exploration of semantic information in the sequence.

Existing studies have shown that category attributes of items or POIs can be used to describe the user's intention [28]. Intuitively, if a user visits a supermarket, his/her intention is shopping. Therefore, we used the category attribute of the POI to represent the user's check-in intention. In addition, the hierarchy of the POI category attribute needs to be considered when mining user's travel intention. Hierarchy refers to the hierarchical dependency between POI category attributes. Figure 2 shows a two-level category hierarchy. The first level includes 9 categories: outdoors and recreation, residence, travel and transport, art and entertainment, college and university, nightlife spot, shop, food, professional spot. The second floor contains 252 categories such as art gallery, movie theater, and subway stations, each of which belongs to a certain first-level category. For example, bus station and subway station all belong to the transportation category.

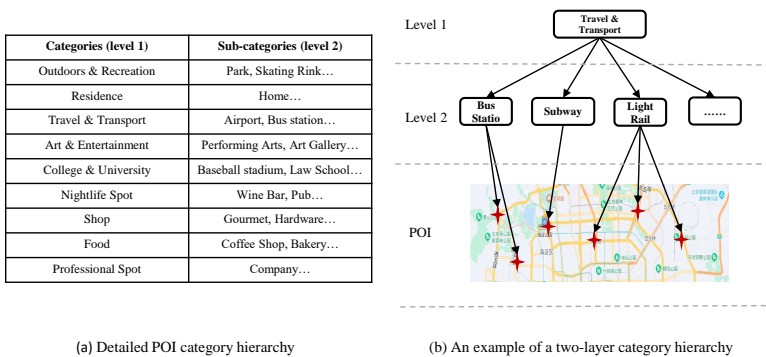

(a) Detailed POI category hierarchy          (b) An example of a two-layer category hierarchy

**Figure 2.** Hierarchical representation of POI categories.

### 3.2. Preliminaries

This section defines the concepts and symbols involved in this paper and formally defines the research questions of this paper. Let $U = \{u_1, u_2, \ldots, u_{|U|}\}$ be a set of users, $L = \{l_1, l_2, \ldots, l_{|L|}\}$ be a set of POIs, and $C = \{c_1, c_2, \ldots, c_{|C|}\}$ be a set of POI categories (e.g., restaurant, fitness center, etc.), where $|U|$, $|L|$, and $|C|$ are the total number of users, POIs, and categories, respectively.

**Definition 1** (POI). *A POI is defined as a spatial location that can be uniquely determined by geographic location information coordinates (longitude and latitude), denoted as $< l_i, lat_i, lon_i >$.*

**Definition 2** (Check-in or visit). *A check-in or visit is a triple $< u_i, l_i, t_i >$, which describes user $u_i \in U$ visiting POI $l_i \in L$ at time $t_i$. Further, the set of historical check-in records of user u can be expressed as: $D_u = \{< u, l_1, t_1 >, < u, l_2, t_2 >, \ldots, < u, l_n, t_n >\}$.*

**Definition 3** (Check-in sequence). *The check-in sequence of user u is his/her historical check-in records in chronological order, denoted by $Q^u = \{q_1^u, q_2^u, \ldots, q_{|Q^u|}^u\}$, in which $q_i^u =< u, l_i, t_i >$ and $\forall i < j, q_i^u.t \le q_j^u.t$.*

**Definition 4** (Semantic sequence)**.** *The semantic sequence is a semantic description of the user's check-in sequence. In this paper, it is composed of the POI category in the check-in sequence and preserves the chronological order. In particular, since the POI categories have a hierarchical structure, users' semantic sequences are also hierarchical. Therefore, the user $u$'s $j$-th semantic sequence can be expressed as: $S_j^u = \{s_1^u, s_2^u, \ldots, s_{|S_j^u|}^u\}$, in which $s_i^u$ denotes the POI category of user $u$'s $i$-th check-in, and it can be denoted as: $s_i^u = <u, c_i^j, t_i>$, in which $c_i^j$ denotes the $j$-th POI category.*

Given a user $u \in U$, his/her check-in sequence $Q^u = \{q_1^u, q_2^u, \ldots, q_t^u\}$, and his/her semantic sequence $S^u = \{S_1^u, S_2^u, \ldots, S_t^u\}$, our goal was to recommend the POIs that $u$ may visit at the next time step $t+1$ (the results are given in the form of a top-$N$ prediction list).

## 4. The Proposed Framework

The structure of the MSAN is shown in Figure 3. The MSAN constructs a multi-task learning framework to predict the user's visiting intention and the next POI to visit separately. In the visiting-intention-prediction task, the hierarchical semantic encoder (HSE) is used to learn the dense representations of hierarchical POI categories and the temporal effect in the semantic sequences. Secondly, the self-attention-based hierarchical intention-aware module (HIAM) was designed to aggregate relevant POI categories within the user's semantic sequence to update the intention representation of each check-in; finally, all candidate categories are matched and sorted by the matching function, and then, a list of the top-$N$ intention predictions is given. In the next POI recommendation task, the spatial–temporal encoder (STE) was designed to learn the dense representations of the POI and spatial–temporal effects within the check-in sequence; secondly, the self-attention-based spatial–temporal-aware module (STAM) is proposed to aggregate the relevant POIs within the user's check-in sequence to update the representation of each check-in; then, the spatial–temporal sequence feature vector and the hierarchical intention feature vector are concatenated as the weighted check-in representations of the spatial–temporal and semantic patterns; finally, the most-plausible candidates from the weighted representations are recalled, and the top-$N$ list of POIs is given. The two learning tasks improve the predictive performance and generalization ability of the model by sharing the intent features of the embedding layer and establishing a weighted loss function for joint learning.

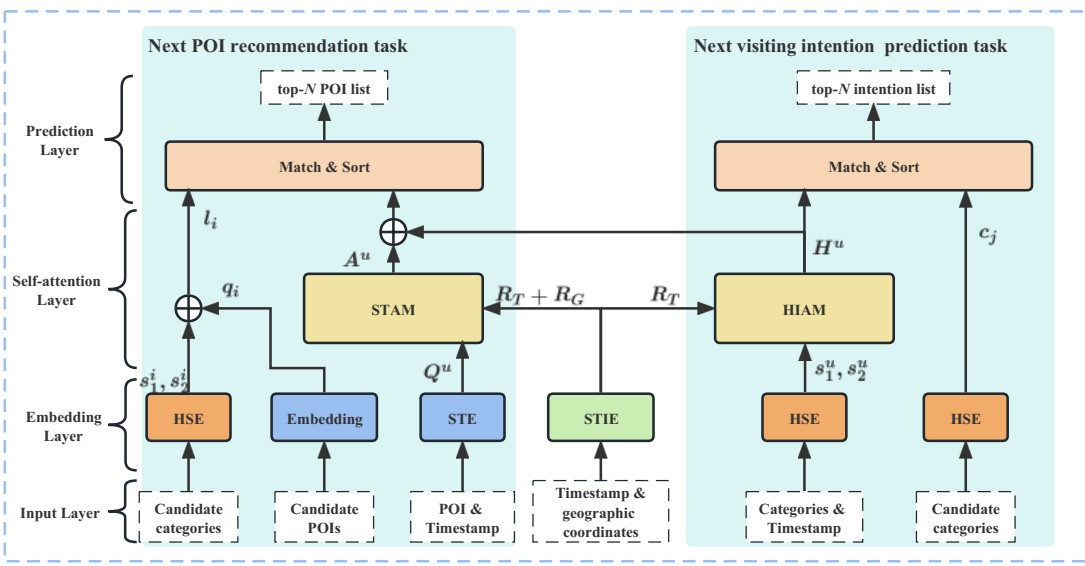

**Figure 3.** The architecture of the proposed MSAN model.

### 4.1. Spatial-Temporal Interval Encoder

Before formally introducing the internal structure of the MSAN, this subsection introduces the spatial–temporal interval encoder (STIE) module. The STIE is used to calculate the time intervals and geographical distances between any check-in behaviors within the user's check-in sequence. Specifically, the STIE calculates the time intervals between the $i$-th and $j$-th visits by $\Delta_{i,j}^t = |t_i - t_j|$, in which $|\cdot|$ denotes the absolute value function. In addition, the STIE calculates the geographical distances between any POIs within the user's check-in sequence by $\Delta_{i,j}^s = Haversine(l_i, l_j)$. If we learn an embedding vector for each time interval and geographic distance, it will lead to a serious data sparsity problem. Inspired by the STRNN model, this paper designed an interpolation embedding layer to embed the spatial and temporal interval information by setting the upper bound unit embedding vector and lower bound unit embedding vector to approximate other interval information, which is calculated as Equation (1):

$$
\begin{aligned}
\mathbf{e}_{i,j}^{\Delta t} &= \frac{\mathbf{e}_{\Delta t}^{sup}(Up(\Delta t) - \Delta t) + \mathbf{e}_{\Delta t}^{inf}(\Delta t - Lo(\Delta t))}{Up(\Delta t) - Lo(\Delta t)} \\
\mathbf{e}_{i,j}^{\Delta s} &= \frac{\mathbf{e}_{\Delta s}^{sup}(Up(\Delta s) - \Delta s) + \mathbf{e}_{\Delta s}^{inf}(\Delta s - Lo(\Delta s))}{Up(\Delta s) - Lo(\Delta s)}
\end{aligned}
\tag{1}
$$

where $\mathbf{e}_{\Delta t}^{sup}, \mathbf{e}_{\Delta t}^{inf} \in \mathbb{R}^d$ represent the time upper and lower bound unit embedding vectors and $\mathbf{e}_{\Delta s}^{sup}, \mathbf{e}_{\Delta s}^{inf} \in \mathbb{R}^d$ represent the spatial upper and lower bound unit embedding vectors, respectively. $Up(\cdot)$ and $Lo(\cdot)$ represent the maximum and minimum values of the time intervals and geographical distances, respectively. Finally, the temporal and spatial relation matrices $\mathbf{R}_T, \mathbf{R}_G$ of user $u$'s check-in sequence are calculated as:

$$
\mathbf{R}_T = \begin{bmatrix} e_{1,1}^{\Delta t} & \cdots & 0 \\ \vdots & \ddots & \vdots \\ e_{n,1}^{\Delta t} & \cdots & e_{n,n}^{\Delta t} \end{bmatrix}
$$

$$
\mathbf{R}_G = \begin{bmatrix} e_{1,1}^{\Delta s} & \cdots & 0 \\ \vdots & \ddots & \vdots \\ e_{n,1}^{\Delta s} & \cdots & e_{n,n}^{\Delta s} \end{bmatrix}
$$

Since only the visit behavior of the first $i$-1 time steps is considered when predicting the POI of the $i$-th time step, the upper right element values of the spatial–temporal relation matrices $\mathbf{R}_T$ and $\mathbf{R}_G$ were set as 0.

### 4.2. Visiting Intention Prediction Task

The visiting intention prediction aims to mine the user's historical semantic sequence and then predict the user's next visiting intention. This paper used the category attribute of the POI to describe the user's intention and predicts the user's next visiting intention by capturing the hierarchical categorical features and temporal effect. Firstly, a hierarchical semantic encoder (HSE) was designed to learn the dense representations of the hierarchical POI categories and temporal effect in the semantic sequences; secondly, a self-attention-based hierarchical intention-aware module (HIAM) was designed to aggregate relevant POI categories within the user's semantic sequence to update the intention representation of each check-in; finally, all candidate categories are matched and sorted by the matching function, and then, a list of the top-$N$ intention predictions is given. The next two subsections introduce the HSE and HIAM modules respectively.

#### 4.2.1. Hierarchical Semantic Encoder

The structure of the HSE module is shown in Figure 4. The HSE takes each item $s_i^u$ in the semantic trajectory of user $u$ as the input and outputs the embedding representation vector $\mathbf{s}_i^u \in \mathbb{R}^d$ of the item. In order to capture the weekly temporal regularity of the users' visiting intentions, we divided the continuous timestamp by $7 \times 24 = 168$ h, which represents the exact hour in a week. Then, for each item $s_i^u = < c_i^j, t_i >$ in the semantic sequence, the HSE maps the check-in timestamp $t_i$ to the corresponding time slice $T_i$, and $s_i^u$ can be further expressed as $s_i^u = < c_i^j, T_i >$. Next, we map the categories $c_i^j$ and $T_i$ to the low-dimensional latent space through the embedding layer. For example, the embedding vector of time slice $\mathbf{T}_i$ is calculated as Equation (2):

$$\mathbf{T}_i = \mathbf{E}_T \cdot \mathbf{o}_{T_i} \tag{2}$$

where $\mathbf{o}_{T_i}$ denotes the one-hot representation of $T_i$, $\mathbf{E}_T$ represents the embedding matrix of the temporal feature, and $\mathbf{T}_i \in \mathbb{R}^d$ is the final embedding vector of the time slice $T_i$.

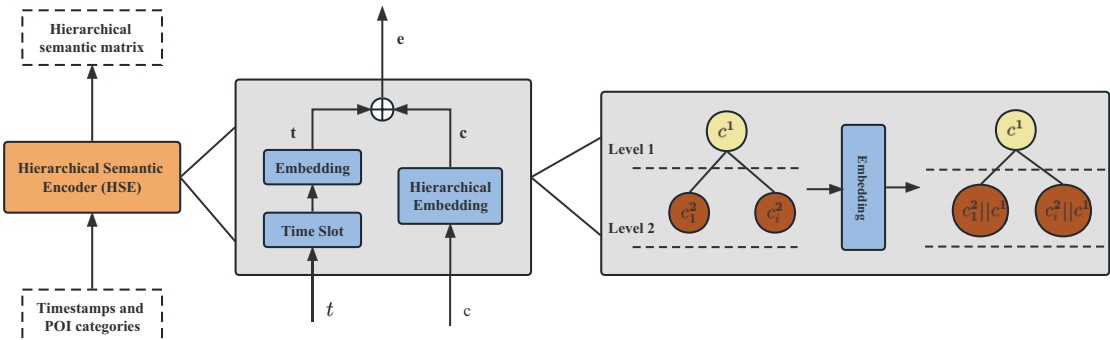

**Figure 4.** The structure of the HSE module.

Different from the embedding process of the above time slices, since the POI categories have a hierarchical structure, it is necessary to model the category hierarchy. Specifically, the parent category can directly use the embedding layer to transform its one-hot vector into the low-dimensional latent space. For the embedding method of the subcategory, this paper concatenates the embedding vector of the subcategory with its parent category embedding vector and uses the concatenated vector as the final embedding of this subcategory, since there is a strong correlation between the subcategories and their parent categories. If $j = 1$, the embedding vector $c_i^1 \in \mathbb{R}^d$ is obtained directly through the embedding layer. If $j = 2$, $c_i^2 = c_i^2 || c_i^1$, where $||$ represents the vector concatenation operation.

Finally, the HSE aggregates the category embedding vector and the time slice embedding vector to calculate the final embedding representation of the current check-in behavior, as shown in Equation (3).

$$\mathbf{e}_i^u = \mathbf{W}_t \mathbf{T}_i + \mathbf{W}_c c_i^j + \mathbf{b} \tag{3}$$

where $\mathbf{T}_i$, $\mathbf{c}_i^j$ are the time slice embedding vector and category embedding vector of the $i$-th item within the user's semantic sequence, $\mathbf{W}_t, \mathbf{W}_c \in \mathbb{R}^{d \times d}$ are the weight parameters, and $b \in \mathbb{R}^d$ is the bias parameter. In addition, since the length of the user's semantic sequence is not fixed, we set a fixed sequence length $n$. If the user's semantic sequence length is greater than $n$, only the last $n$ records in the sequence will be considered. On the contrary, if the user's semantic sequence length is less than $n$, the left of this sequence will be padded zero vectors. When the model is updated through the forward propagation process, these zero vectors will be masked off and do not participate in the learning process. Finally, the $j$-th layer semantic sequence embedding vector of user $u$ is $\mathbf{S}_j^u = \{\mathbf{e}_1^u, \mathbf{e}_2^u, \cdots, \mathbf{e}_n^u\}$.

### 4.2.2. Hierarchical Intention-Aware Module

In order to capture the hierarchical features of the user's visiting intention, the HIAM designs two stacked interval-aware encoders to mine the user's intention from two different granularities, which is shown in Figure 5.

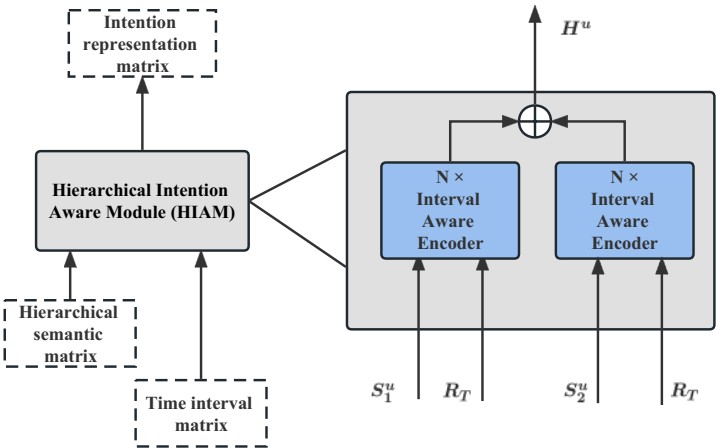

**Figure 5.** The structure of hierarchical intention-aware module.

The interval-aware encoder takes user $u$'s semantic sequence matrix $\mathbf{S}_j^u$ and temporal relation matrix $\mathbf{R}_T$ as the input and takes the attention encoding matrix $\mathbf{A}^u$ as the output. The structure of the interval-aware encoder is shown in Figure 6. Firstly, the interval-aware attention layer uses three linear layers to transform the semantic sequence matrix $\mathbf{S}_j^u$ of user $u$ into $\mathbf{Q}$(Query), $\mathbf{K}$(Key), and $\mathbf{V}$(Value), respectively, which are calculated as Equation (4):

$$
\begin{aligned}
\mathbf{Q} &= \mathbf{S}_j^u \mathbf{W}_Q \\
\mathbf{K} &= \mathbf{S}_j^u \mathbf{W}_K \\
\mathbf{V} &= \mathbf{S}_j^u \mathbf{W}_V
\end{aligned}
\tag{4}
$$

where $\mathbf{W}_Q, \mathbf{W}_K, \mathbf{W}_V \in \mathbb{R}^{d \times d}$ are the weight parameters and $\mathbf{Q}, \mathbf{K}, \mathbf{V} \in \mathbb{R}^{n \times d}$ are the updated representation matrices. Then, the time relation matrix $\mathbf{R}_T$ is introduced into the self-attention layer in the form of matrix elementwise addition (Equation (5)):

$$
\mathbf{A}^u = \left(\mathbf{M} * softmax\left(\frac{\mathbf{Q}\mathbf{K}^T + \mathbf{R}_T}{\sqrt{d}}\right)\right)\mathbf{V}
\tag{5}
$$

where $\mathbf{M} \in \mathbb{R}^{n \times n}$ is the mask matrix whose elements above the diagonal are filled with $-\infty$. $\frac{\mathbf{Q}\mathbf{K}^T + \mathbf{R}_T}{\sqrt{d}}$ means that the serial correlations are continuously corrected through the time interval relation, so that the self-attention mechanism can mine the temporal correlation in the sequence. $\mathbf{A}^u \in \mathbb{R}^{n \times d}$ is the calculated attention weight matrix.

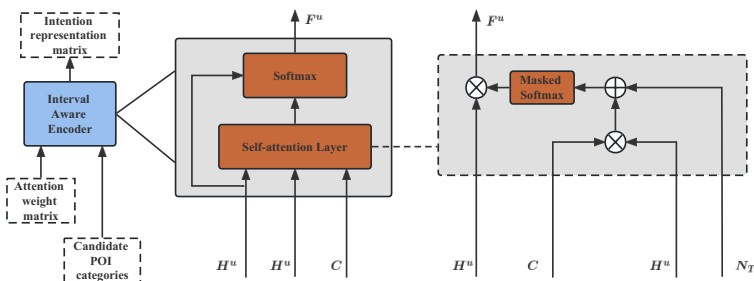

**Figure 6.** The structure of interval aware encoder.

As mentioned above, the interval-aware self-attention layer combines all semantic sequence items, relative time intervals, and adaptive weights together based on a linear combination. To optimize the attention weight matrix, we introduced the pointwise feed-forward network (FFN), which is calculated as Equation (6):

$$\mathbf{H}^u = ReLU(0, \mathbf{A}^u \mathbf{W}_1 + \mathbf{b}_1)\mathbf{W}_2 + \mathbf{b}_2 \tag{6}$$

where $\mathbf{W}_1, \mathbf{W}_2 \in \mathbb{R}^{d \times d}$ are the weight parameters, $\mathbf{b}_1, \mathbf{b}_2 \in \mathbb{R}^d$ are the bias parameters, and $\mathbf{H}^u \in \mathbb{R}^{n \times d}$ is the optimized attention weight matrix.

Studies have shown that stacking multi-layer neural networks in the model will cause errors to accumulate continuously, resulting in unstable model training performance (gradient disappearance) [29]. Therefore, we introduced the residual network [30] and layer normalization [31]. Taking the input vector **x** as an example, the calculation equations are shown in (7):

$$\mathbf{x} = \mathbf{x} + Layer(LayerNorm(\mathbf{x}))$$
$$LayerNorm(\mathbf{x}) = \alpha \odot \frac{\mathbf{x} - \mu}{\sqrt{\sigma^2 + \epsilon}} \tag{7}$$

where $\odot$ represents that the corresponding positions of the two matrices are multiplied, $\alpha, \gamma$, and $\epsilon$ are learned parameters for controlling numerical scaling and biasing, and $\mu$, $\sigma$ represent the mean and variance of each dimension of the vector. Based on the above method, we can obtain a two-level user's intent representation matrix $\mathbf{H}_1^u$ and $\mathbf{H}_2^u$. Then, the two matrices are fused by the vector-by-vector addition operation, and the user's final intention representation matrix is calculated as Equation (8):

$$\mathbf{H}^u = \mathbf{H}_1^u \oplus \mathbf{H}_2^u \tag{8}$$

where $\oplus$ represents the vector addition operation and $\mathbf{H}^u \in \mathbb{R}^{n \times d}$ denotes the final intent representation matrix.

### 4.2.3. Intention Prediction

Given the updated intention representation matrix $\mathbf{H}^u \in \mathbb{R}^{n \times d}$ of user $u$, this paper used the latent factor model to calculate the probability that the visiting intention of user $u$ at time step $i + 1$ is $c_j$, and the equation is shown in (9):

$$R_{i,c_j}^u = \mathbf{H}_i^u \cdot \mathbf{c}_j \tag{9}$$

where $\mathbf{c}_j \in \mathbb{R}^d$ is the embedding vector of the candidate POI category.

In this paper, the binary cross-entropy loss function was used, and the model was optimized with the goal of minimizing the loss. The calculation equation is shown in (10).

$$\mathcal{J}_c = -\sum_{S^u \in \mathcal{S}} \sum_{t \in [1,2,\cdots,n]} (log\sigma(R_{t,o_t}^u) + \sum_{\overline{o_t} \notin S_u} log(1 - \sigma(R_{t,\overline{o_t}}^u))) \tag{10}$$

where $\mathcal{S}$ denotes the set of all trained semantic sequences and $\sigma(x) = \frac{1}{1+e^{-x}}$ is the softmax function. During the training process, for each positive class sample $o_t$ in the sequence, we randomly sampled a negative class sample $\overline{o_t}$ from the POIs that the user had not visited and used the Adam optimizer [32] to optimize the parameters in the model.

### 4.3. Next POI Recommendation Task

The next POI recommendation aims to predict POIs that users may visit at the next time step. Firstly, this paper designed a spatial–temporal sequence encoder (STE) to embed the POI and spatial–temporal effects within the check-in sequence into the latent space; secondly, a spatial–temporal-aware module (STAM) is proposed to capture the spatial–temporal sequential patterns in the user's check-in sequence; then, the spatial–temporal sequence feature vector and the hierarchical intention feature vector are concatenated as the

weighted check-in representations of the spatial–temporal and semantic patterns; finally, the most-plausible candidates from the weighted representations are recalled, and the top-*N* list of POIs is given. The next two subsections introduce the STE and STAM, respectively.

### 4.3.1. Spatial-Temporal Sequence Encoder

Given the check-in trajectory $Q^u = \{q_1^u, q_2^u, \ldots, q_{|Q^u|}^u\}$ of user *u*, the STE takes each item in the user's check-in trajectory as the input and takes the embedding representation vector $q_i^u \in \mathbb{R}^d$ of this item as the output. Similar to the embedding of POI categories in the HSE, the POI and timestamp embeddings of each check-in item are denoted as $l_i$ and $T_i$, respectively. Furthermore, the geography encoder [3] is introduced to embed the exact position of the POI by first mapping the latitude and longitude into a grid and then encoding the unique ID (quadkey) of the grid with a self-attention network. Thus, we denote the embedded representations of the POIs' positions as $P_L = [\mathbf{p}_1^l, \mathbf{p}_2^l, \cdots, \mathbf{p}_L^l]$. Hence, the spatial–temporal-aware visiting aggregation $\mathbf{q}_i^u$ is represented by Equation (11):

$$\mathbf{q}_i^u = \mathbf{W}_t \mathbf{T}_i + \mathbf{W}_l \mathbf{l}_i + \mathbf{p}_i^l + \mathbf{b} \tag{11}$$

where $\mathbf{W}_t, \mathbf{W}_l \in \mathbb{R}^{d \times d}$ are the weight parameters $\mathbf{b} \in \mathbb{R}^d$ is the bias parameter. In addition, since the length of the user's semantic sequence is not fixed, we set a fixed sequence length *n*. If the user's semantic sequence length is greater than *n*, only the last *n* records in the sequence will be considered. On the contrary, if the user's check-in sequence length is less than *n*, the left of this sequence will be padded zero vectors. When the model is updated through the forward propagation process, these zero vectors will be masked off and do not participate in the learning process. Finally, the check-in sequence embedded representations are denoted as $\mathbf{q}^u = \{\mathbf{q}_1^u, \mathbf{q}_2^u, \cdots, \mathbf{q}_n^u\}$.

### 4.3.2. Spatial-Temporal Aware Module

The STAM consists of *N* stacked interval-aware encoders, each of which takes the spatial–temporal sequence representation $\mathbf{q}^u \in \mathbb{R}^{n \times d}$ and the corresponding spatial–temporal relation matrix $\mathbf{R}_{TG} \in \mathbb{R}^{n \times n}$ as the input and takes the updated sequence representation matrix $\mathbf{A}^u \in \mathbb{R}^{n \times d}$ as the output. The spatial–temporal relation matrix $\mathbf{R}_{TG}$ is obtained by adding $\mathbf{R}_T$ and $\mathbf{R}_G$ item by item, which is denoted as: $\mathbf{R}_{TG} = \mathbf{R}_T + \mathbf{R}_G$. The structure of the interval-aware encoder is introduced in Section 4.2.2.

### 4.3.3. Next POI Recommendation

Given the updated spatial–temporal sequence representation matrix $\mathbf{A}^u \in \mathbb{R}^{n \times d}$ of user *u*, this paper used the latent factor model to calculate the probability that user *u* will visit POI $l_j$ at time step $i + 1$, and the equation is shown in (12):

$$R_{i,l_j}^u = \mathbf{A}_i^u \cdot \mathbf{l}_j \tag{12}$$

where $\mathbf{l}_j \in \mathbb{R}^d$ is the embedding vector of candidate POI $l_j$.

Sequential recommendation models usually use the binary cross-entropy loss function. During training, most of the models randomly select an unvisited item as a negative sample for each positive sample, which makes it impossible to effectively use a large number of negative samples. In view of this, this paper adopted the negative sample sampling method [4] and tuned the number of negative samples by setting the parameter $\phi$. The calculation equation is shown in (13):

$$\mathcal{J}_l = -\sum_{Q^u \in \mathcal{Q}} \sum_{t \in [1,2,\cdots,n]} \left( log\sigma(R_{t,l_t}^u) + \sum_{\overline{l_t} \notin Q^u} log(1 - \sigma(R_{t,\overline{l_t}}^u)) \right) \tag{13}$$

where $\mathcal{Q}$ represents the set of check-in sequences and $\sigma(x) = \frac{1}{1+e^{-x}}$ is the softmax function. During training, for each positive sample $l_t$ in the sequence, we randomly sampled $\phi$ nega-

tive samples from the POIs that the user had not visited and used the Adam optimizer [32] to optimize the parameters in the model.

*4.4. Model Training*

Both of the above two tasks use binary cross-entropy loss as the loss function of the model. This paper proposes a multi-task learning strategy to jointly optimize the loss of the two tasks. The final loss function calculation formula of the model is calculated as (14):

$$\mathcal{J} = \lambda \mathcal{J}_c + (1 - \lambda)\mathcal{J}_l \tag{14}$$

where $\lambda$ denotes weight parameter.

**5. Experiment**

*5.1. Datasets*

In order to verify the effectiveness of the MSAN model, this paper used two real-world LBSNs datasets, NYC and TKY [33], for the experiments. The statistics of the datasets are listed in Table 1. Both datasets include about 10 months from April 2012 to February 2013. The attributes of each check-in record include the user ID, the POI ID, the check-in timestamp, the geographic information of the POI (including latitude and longitude information), and the POI category information. Given the check-in trajectory $Q^u = \{q_1^u, q_2^u, \cdots, q_n^u\}$ of each user $u$, we divided it into the training set, validation set, and test set, respectively. Specifically, the training set included $n - 3$ check-in sequences, with the first $[1, n - 3]$ check-ins in the sequence as the input sequence and the $[2, n - 2]$-th check-in as the label, respectively; the verification set used the first $n - 2$ check-ins as the input sequence and used the $(n - 1)$-th check-in as the label; the test set used the first $n - 1$ check-ins in the sequence as the input sequence, and the $n$-th check-in was used as the label.

**Table 1.** Statistics of the datasets.

| Dataset | # of Users | # of POIs | # of Check-Ins | Avg. # of Check-In |
|---------|-----------|-----------|----------------|--------------------|
| NYC | 1083 | 38,333 | 227,428 | 210.0 |
| TKY | 2293 | 61,858 | 573,703 | 250.2 |

*5.2. Evaluation Metrics*

We adopted two widely used metrics, Acc@N and NDCG@N, to evaluate the performance of the MSAN model. Acc@N measures the percentage of recommended top-N POIs that are actually visited by the user, which is calculated as Equation (15):

$$Acc@N = \frac{\sum_{i=1}^{L} \#hit_i@N}{|L|} \tag{15}$$

where $\#hit_i@N$ represents the number of POIs that are actually visited by the user and appear in the top-N recommended list. If the top-N recommended list includes $i$, then $\#hit_i@N = 1$, else $\#hit_i@N = 0$. $|L|$ represents the total number in the test set.

NDCG@N measures the quality of top-N ranking list, and it is calculated as Equation (16):

$$NDCG@N = \frac{DCG@N}{IDCG@N}$$
$$DCG@N = \sum_{i=1}^{N} \frac{2^{rel_i} - 1}{log_2(i + 1)} \tag{16}$$

where $n$ is the ranking of the POIs actually visited by the user in the recommended list and $rel_i$ represents the rank correlation of each candidate POI between the recommended

list and the test list. In this paper, if the ranked $i$ POI is in the test list, then $rel_i = 1$, else $rel_i = 0$. IDCG@$N$ is the maximum value of DCG@$N$.

*5.3. Baseline Models*

In order to verify the effectiveness of the MSAN model, this paper selected the following baseline models to compare with the MSAN:

- STRNN [2]: The STRNN is an improvement to the traditional RNN, which incorporates the spatial–temporal interval matrix between consecutive check-in behaviors into the RNN to model spatial–temporal context information.
- HST-LSTM [25]: HST-LSTM integrates spatial–temporal context information into LSTM and proposes a trajectory hierarchy division method based on functional areas, which alleviates the problem of data sparsity by mining the spatial–temporal transition patterns between different functional areas.
- TMCA [6]: TMCA designs an LSTM-based encoder–decoder network and proposes three types of attention mechanisms to fuse the spatial–temporal context information, including multi-level context attention (micro and macro levels) and temporal attention.
- LSTPM [5]: LSTPM divides the user's historical check-in trajectory into long-term trajectory sequences and short-term trajectory sequences and designs corresponding LSTM networks to capture long-term and short-term sequence preferences, respectively.
- iMTL [34]: iMTL is an interactive multi-task learning framework, which uses a dual-channel encoder based on an LSTM network to capture the temporal features and POI categorical features separately, and it designs a task-specific decoder to optimize the interactive learning of the two tasks.
- GeoSAN [3]: GeoSAN is a self-attention-based sequence prediction model that can capture long-term sequence dependencies, and a self-attention-based geography encoder is proposed to capture the spatial correlations between POIs within the user's check-in trajectory.
- SANST [7]: SANST uses the self-attention network (SAN) to simultaneously fuse spatio-temporal context information. Specifically, it uses the hierarchical grid embedding method to capture geographic clustering features.
- STAN [4]: STAN proposes to model the spatial–temporal intervals between non-adjacent POIs and non-sequential check-ins and integrate them into the self-attention mechanism.
- CHA [35]: CHA proposes to explore the category hierarchy of POIs to help learn robust location representations even when there is insufficient data. Moreover, it develops a spatial–temporal decay LSTM to model the influence of the time interval and distance and proposes a discrete Fourier-series-based periodic attention to model users' innate periodic activities.
- LSMA [36]: LSMA utilizes a multi-level attention mechanism to study the multi-factor dynamic representation of a user's check-in behavior and non-linear dependence between check-ins in his/her check-in trajectory. Moreover, it combines the long- and short-term preferences of the user to form the final user preference.

*5.4. Experimental Setting*

Both the spatial–temporal sequence length and the semantic sequence length in the model were set to 100. All embedding vectors in the NYC and TKY datasets have dimension 50 in the MSAN. We adopted the number of negative samples $\phi = 10$ in Section 4.3.3 to train the MSAN model as the experiments of previous work showed that this is the optimal setting [4]. In addition, this paper used the Adam optimization algorithm [32] to tune the parameters, and the learning rate was set to 0.001. The task weight parameter of the model was $\lambda = 0.7$. The learning rate and embedding dimension were set to 0.001 and 50 for all

baseline models, respectively. The other parameters of the baseline models were set to their default values that came with the original paper.

*5.5. Performance Evaluation*

5.5.1. Overall Performance

Tables 2 and 3 show the comparative experimental results (Acc@*N* and NDCG@*N*) of the MSAN model on the NYC and TKY datasets respectively. The last row indicates the degree of performance improvement of the MSAN model compared to the best-performing baseline model (STAN). Tables 2 and 3 indicate that the MSAN model had superior performance compared with all baseline models on Acc@*N* and NDCG@*N* for the two datasets. Compared with STAN, the performance of the MSAN improved by 6–8% on Acc@*N*, and the *NDCG@N* index improved by 3–9%.

**Table 2.** Performance comparison of all models under NYC.

| Model | Acc@10 | Acc@20 | NDCG@10 | NDCG@20 |
|---|---|---|---|---|
| STRNN | 0.153 | 0.192 | 0.103 | 0.175 |
| TMCA | 0.161 | 0.205 | 0.117 | 0.182 |
| LSTPM | 0.169 | 0.213 | 0.128 | 0.197 |
| iMTL | 0.187 | 0.255 | 0.137 | 0.212 |
| HST-LSTM | 0.202 | 0.298 | 0.168 | 0.231 |
| LSMA | 0.213 | 0.305 | 0.188 | 0.247 |
| CHA | 0.219 | 0.323 | 0.199 | 0.257 |
| SANST | 0.227 | 0.342 | 0.205 | 0.263 |
| GeoSAN | 0.276 | 0.367 | 0.228 | 0.284 |
| STAN | 0.304 | 0.393 | 0.236 | 0.291 |
| MSAN | 0.324 | 0.425 | 0.251 | 0.31 |
| **Improv.** | **6.6%** | **8.1%** | **6.4%** | **6.5%** |

**Table 3.** Performance comparison of all models under TKY.

| Model | Acc@10 | Acc@20 | NDCG@10 | NDCG@20 |
|---|---|---|---|---|
| STRNN | 0.147 | 0.188 | 0.133 | 0.157 |
| TMCA | 0.153 | 0.196 | 0.146 | 0.178 |
| LSTPM | 0.188 | 0.239 | 0.158 | 0.201 |
| iMTL | 0.205 | 0.258 | 0.164 | 0.235 |
| HST-LSTM | 0.217 | 0.276 | 0.173 | 0.251 |
| LSMA | 0.225 | 0.283 | 0.177 | 0.259 |
| CHA | 0.231 | 0.286 | 0.181 | 0.264 |
| SANST | 0.243 | 0.292 | 0.188 | 0.265 |
| GeoSAN | 0.256 | 0.314 | 0.207 | 0.271 |
| STAN | 0.297 | 0.325 | 0.227 | 0.282 |
| MSAN | 0.317 | 0.355 | 0.234 | 0.291 |
| **Improv.** | **6.7%** | **9.2%** | **3.1%** | **3.2%** |

Among all the comparison models, the best performance was based on the self-attention mechanism, such as SANST, GeoSAN, and STAN, which proved that the self-attention mechanism has a significant advantage in sequence modeling. In addition, the performance of the STAN model was better than that of the GeoSAN model, because the GeoSAN model only models the geographical influence factors and does not consider the time interval characteristics within the user's check-in sequence, while STAN uses the spatial interval matrix instead of GeoSAN's geographic grid embedding, which can capture geographic information more accurately. In all comparison models based on the RNN and LSTM, the performance of the LSMA and CHA models was the best, and there was not much difference in the prediction performance between the two models. The reason is that

LSMA captures the category transition patterns in users' check-in trajectories and takes into account the differences in users' long-term and short-term preferences. The performance of CHA was slightly higher than that of LSMA, mainly because it considers the hierarchical features of POI categories and designs an encoding method to model this, which proves that capturing the hierarchical features of POI categories is beneficial for mining user's visiting preference. The performance of the TMCA, LSTPM, iMTL, and HST-LSTM models was better than the STRNN, which proved that LSTM is stronger than RNN in terms of long sequence modeling ability. In addition, the LSTPM model considers the difference of users' long-term and short-term preferences when modeling and designs different schemes to capture the long-term and short-term preferences of users' check-in behavior, as well as uses the attention mechanism when modeling long-term preferences, so its performance was better than TMCA. The performance of the iMLT and HST-LSTM models was superior. iMTL introduces the semantic information of users' check-in behavior and uses a multi-task learning framework to take the semantic sequence mining task as the auxiliary task of temporal feature mining. The experimental results showed that the joint learning between the two tasks helped each other, which proved the validity of the multi-task learning framework to learn the semantic sequence and spatial–temporal sequence of the user's check-in behavior respectively. HST-LSTM incorporates spatial–temporal interval features within the user's check-in sequences into LSTM and showed the best performance among the LSTM-based models, which fully demonstrated the effectiveness of spatial–temporal interval features for modeling user's check-in behavior. The MSAN model integrates the advantages displayed by the iMTL and HST-LSTM models. It mines user's visiting intention from semantic sequences and uses the user's visiting intention prediction task as the auxiliary task to predict the next POI, which greatly improved the recommendation accuracy.

### 5.5.2. Ablation Study

In order to further prove the effectiveness of each module of the MSAN model, this section designs the variants of the MSAN model to conduct experiments to quantitatively analyze the key parts of the model:

1.  MSAN-HIAM: We removed the HIAM module and only used the self-attention mechanism to capture the sequential correlations within the user's check-in trajectory. Correspondingly, Equation (8) was modified as: $\mathbf{H}^u = \mathbf{H}_1^u$.
2.  MSAN-Intention: The visiting intention prediction module was removed, and we kept only the next POI recommendation.

Taking the NYC dataset as an example, we experimented with the performance of different variants of the models, and the results are shown in Figure 7. Figure 7 indicates that the performance of the MSAN model was always better than other variants of the models under the Acc@*N* and NDCG@*N* evaluation indicators. Specifically, the MSAN was 5–19% higher than the MSAN-Intention model on Acc@*N* and 18–33% higher than the MSAN-Intention model on NDCG@*N*, which demonstrated the effectiveness of using user's visiting intention prediction as the auxiliary task for the next POI recommendation. In addition, it can also be seen from the figure that the performance of the MSAN model was always better than that of the MSAN-HIAM model, indicating that hierarchical modeling of users' semantic sequences has a significant impact on improving model performance.

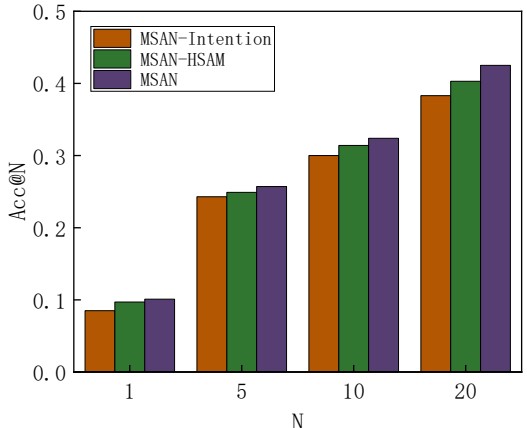 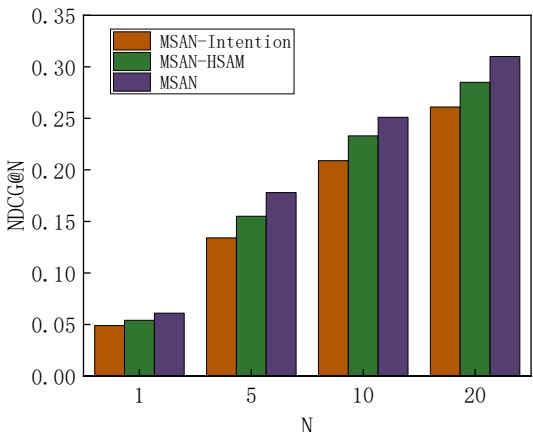

(a) Acc@N of different variant models under NYC  (b) Acc@N of different variant models under NYC

**Figure 7.** Performance comparison of different variants of the models.

### 5.5.3. Stability Study

**Effect of task weight parameter** $\lambda$**:** The interactive multi-task learning model tends to be biased towards specific tasks, so an appropriate weight parameter $\lambda$ needs to be selected experimentally. We varied the value of the weight parameter $\lambda$ from 0.1 to 1.0 with a step of 0.1, and the experimental results are shown in Figure 8. It can be seen from the figure that, when $\lambda = 0.7$, the model achieved the best performance on both datasets. At this time, the POI prediction task was the dominant task, and the intention prediction task was an auxiliary task. When $\lambda = 1$, the MSAN model completely ignored the user intention prediction task, and its performance dropped significantly. When $\lambda$ was small, the model focused too much on the intention prediction task and failed to capture the spatial–temporal sequential dependencies between POIs, resulting in weaker performance.

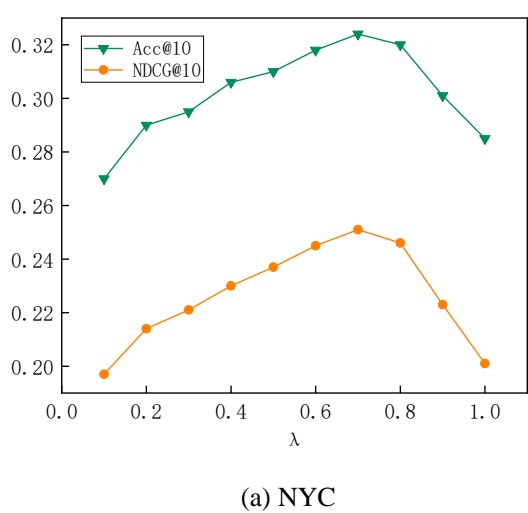 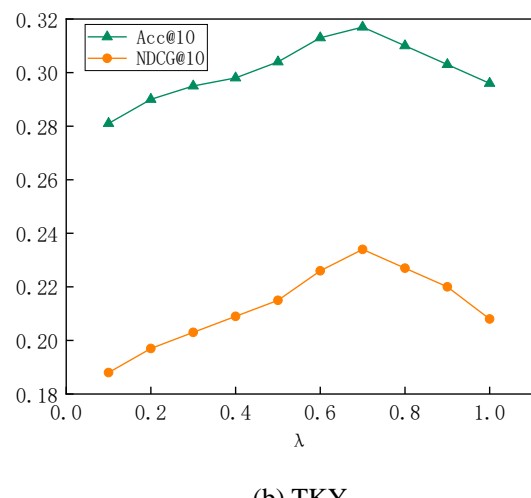

(a) NYC  (b) TKY

**Figure 8.** Effect of parameter $\lambda$.

**Effect of embedding dimension** $d$: This section studies the impact of the embedding vector dimension $d$ of each element in the model. We varied the embedding dimension $d$ to observe the performance of the MSAN model in terms of Acc@10, and the results are shown in Figure 9. It can be seen from the figure that, when d was too small, the performance of the model was relatively poor because it could not effectively extract the features of the corresponding elements. When $d$ exceeded a certain threshold, the performance of the model did not improve significantly, or even there would be a downward trend, making

the model face the risk of overfitting. In this experiment, the threshold was 50, so in the comparison experiment, the dimension of the embedding vector of each element was set to $d = 50$.

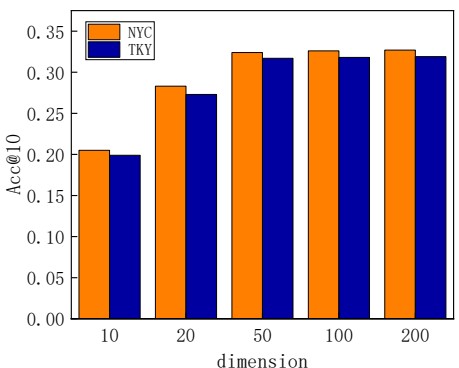
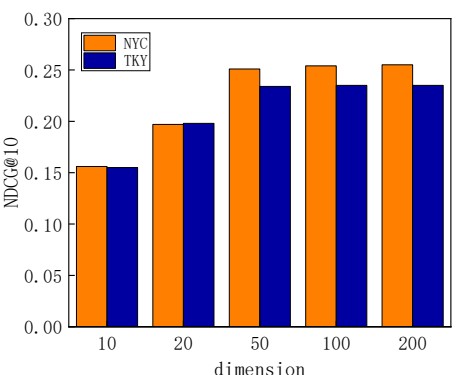

(a) Acc@10 of different dimensions under different datasets  (b) NDCG@10 of different dimensions under different datasets

**Figure 9.** Effect of embedding dimension $d$ of MSAN model.

### 5.5.4. Analysis of Training Process

In order to explore the relationship between the two prediction tasks and their impact on model performance, we analyzed the intent prediction loss and POI prediction loss during training, and the results are shown in Figure 10. It can be seen from the figure that, in the early stage of training, the loss value trend of the intention prediction task was more severe than that of the POI prediction task, and its convergence speed was faster (the intent prediction task started to converge when the number of iterations was about 40, while the POI prediction task was 60), which means that the model paid more attention to the learning of the user's visiting intention in the early stage of training and then optimized the POI prediction task after it converged. This phenomenon further proved the effectiveness of selecting the intention prediction task as an auxiliary task for POI prediction.

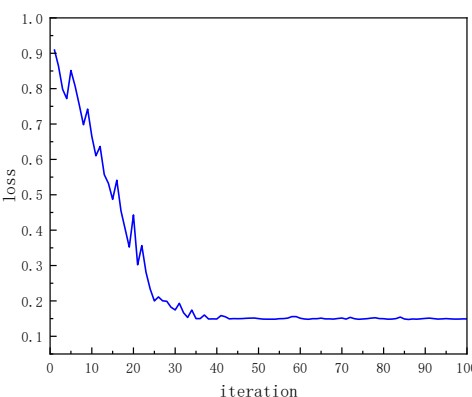
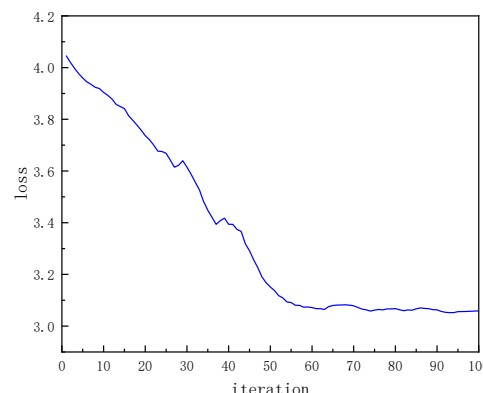

(a) The training loss of Intention Prediction task  (b) The training loss of POI Prediction task

**Figure 10.** The training loss of intention prediction and POI prediction task.

### 5.5.5. Interpretability Study

As mentioned above, the check-in behavior of different users has different spatio-temporal sequence features and semantic sequence features, and the MSAN can adaptively capture the spatio-temporal and semantic patterns of the user's check-in behavior. To understand the mechanism of the MSAN, we randomly selected a user $u$ from the NYC dataset and predicted his/her visiting intention and check-in behavior in the next time step. Firstly,

we separated the two learning tasks of the MSAN framework into two separate modules: the user's visiting intention prediction module and the next POI prediction module. Based on these two modules, we predicted the user's visiting intention and the next possible POI to visit in the next time step, respectively. Secondly, we predicted the next POI that the user may visit based on the MSAN. Finally, we explain the mechanism of the MSAN model by visualizing the attention weights of each POI in the user's check-in sequence, and the results are shown in Figure 11b. Figure 11a shows the distribution of geographic intervals between the last POI visited by user *u* and the POIs visited by user *u* the last 10 times. The geographical distance was relatively close, and the geographical correlation was relatively strong. Figure 11b shows that, under the condition of only considering the next POI prediction task, the module mainly focused on POIs that had a small geographical distance from the user's current location, such as POIs 1, 9, and 10. At this time, the prediction result of the module was $l_{10264}$, which is the company of user *u*. Under the condition of only considering the intention prediction task, this module paid more attention to POIs 4 (art and entertainment), 5 (food), 7 (outdoors and recreation), 9 (shop), and 10 (food) from the trajectory, so it can be known that the visiting intention of user *u* was entertainment and shopping. At this time, the prediction result of this module was an art and entertainment place. The MSAN model not only paid more attention to POIs with small space–time intervals, but also had a high degree of attention to entertainment or shopping POIs. At this time, it was predicted that the POI that the user would visit in next time step was $l_{470}$, which belongs to the visiting intention predicted above, consistent with the actual result. From the above analysis, it can be proven that the travel intention prediction task in the MSAN can effectively promote the self-attention mechanism to consider the user's visiting intention and adjust the corresponding attention weight in the sequence to adaptively capture the spatiotemporal correlation in the user check-in sequence and semantic relevance, thereby improving POI prediction performance.

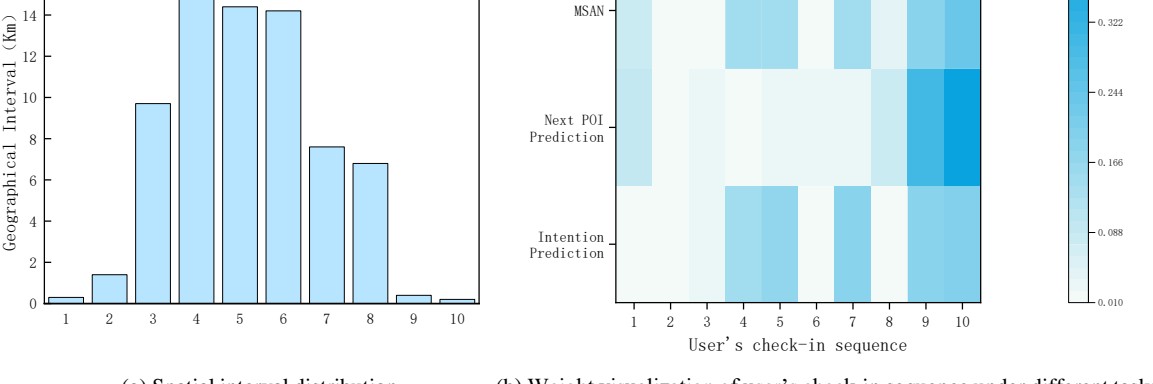

(a) Spatial interval distribution　　(b) Weight visualization of user's check-in sequence under different tasks

**Figure 11.** Interpretability analysis of MSAN.

## 6. Conclusions

In this paper, we proposed a self-attention- and multi-task-based model (MSAN) for next POI recommendation. The MSAN takes the user's visiting intention prediction task as the auxiliary task of the next POI recommendation task and realizes the knowledge sharing between the two tasks by building a multi-task learning framework, which improved the performance and generalization ability of the model. The user's visiting intention prediction task uses hierarchical POI category attributes to describe the user's visiting intention and designs a hierarchical intention embedding method to mine the hierarchical features between intentions of different granularities, then proposes a hierarchical intention-aware module based on a self-attention mechanism to mine the temporal and hierarchical semantic features within the user's check-in sequence. The next POI recommendation task

uses the spatial–temporal-aware module to mine the spatial–temporal sequential patterns within the user's check-in trajectory and fuses this with the hierarchical intention features to generate the next candidate POI list. Experiments based on two real datasets verified the effectiveness of the model.

**Author Contributions:** Conceptualization, Zhiming Ding; methodology, Jing Tian; software, Jing Tian; validation, Zhiming Ding and Zilin Zhao; formal analysis, Zhiming Ding and Zilin Zhao; investigation, Jing Tian; resources, Zhiming Ding; data curation, Zhiming Ding; writing—original draft preparation, Jing Tian; writing—review and editing, Zhiming Ding; visualization, Jing Tian; supervision, Zhiming Ding; project administration, Zhiming Ding; funding acquisition, Zhiming Ding. All authors have read and agreed to the published version of the manuscript.

**Funding:** This work was supported by the National Key R & D Program of China (No.2022YFF0503900) and the Key R & D Program of Shandong Province (No.2021CXGC010104).

**Data Availability Statement:** The data presented in this study are available from http://www-public.imtbs-tsp.eu/~zhang_da/pub/dataset_tsmc2014.zip (accessed on 25 March 2021).

**Conflicts of Interest:** The authors declare no conflict of interest.

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
