# Peer review of "Next Point-of-Interest Recommendation Based on Joint Mining of Spatial–Temporal and Semantic Sequential Patterns"

_ijgi, doi:10.3390/ijgi12070297_

Round 1

Reviewer 1 Report

MSAN, which recommends POI by predicting the user's next visit intention, performs better than other algorithms, as the author proved through experiments. In particular, the significance of the paper could be accurately grasped through the part of the experiment in which the user's visit intention prediction module made in ablation study was removed and only the following POI recommendation was maintained. The overall composition and experimental method of the paper are excellent. However, to increase the paper's completeness, I would like to comment on only the following two things.

First, regarding the baseline models mentioned on page 12 of the paper, I wonder how the parameters were treated when applying the experiment. A reference is attached in the text, but it does not include whether it is used the same as the reference or whether the default value is applied. It is necessary to mention on this.

Second, regarding the experimental results, this paper only lists the overall performance through Table 2 and Table 3. It would be nice to explain the results in more detail for many readers. In addition to overall performance, if you present the results of how input enters through MSAN to predict the user's intention to visit and which POI is recommended, you can improve the reader's understanding of this paper.

Reviewer 2 Report

The paper proposes a self-attention and multi-task based method, called MSAN, to simultaneously explore the spatial-temporal relationships and semantic sequence patterns of user trajectories. The terminology and expressions are used appropriately and the style of the paper is up to standard, but there are some issues.

1. The aesthetics of the method block diagram need to be improved and are difficult to understand.

2. How is the hierarchy divided, and what do the parent and child classes represent, respectively? I didn't find it in your paper, which is important.

3. It is suggested that the diagrams in the paper be further modified; for example, the arrows in Figure 3 are inconsistent in size and thickness and are not centered; Figure 4 has the same problem.

4. The format of the references is not uniform.

5. The comparison experiments are all before 2021; is it possible to add the methods of the last two years?

6. Formatting problems in the text, e.g., the last paragraph lacks a period.

The author needs to make changes to the English grammar in the manuscript.

Reviewer 3 Report

In this manuscript, the authors propose a new method for the next POI recommendation. The proposed method, namely MSAN, jointly models spatial-temporal and semantic sequential patterns to generate more accurate results. Overall, this paper is well written. The contributions and technique novelties are clear. However, the authors should pay attention to some minor issues.

1. Two tasks are employed to yield a better recommendation result. The relationship between the two tasks should be further analyzed. Intuitively, the result of intention prediction would influence the results of POI recommendations. Whether the joint loss in MSAN can model this correlation or not 

2. In figure 7, the text of legend is chinese

The presentation of this manuscript is satisfactory. Some minor typos should be corrected. 

Round 2

Reviewer 2 Report

Minor editing in English is required

Minor editing in English is required